# A Comparative Analysis of Pain Control Methods after Ankle Fracture Surgery with a Peripheral Nerve Block: A Single-Center Randomized Controlled Prospective Study

**DOI:** 10.3390/medicina59071302

**Published:** 2023-07-14

**Authors:** Jeong-Kil Lee, Gi-Soo Lee, Sang-Bum Kim, Chan Kang, Kyong-Sik Kim, Jae-Hwang Song

**Affiliations:** 1Department of Orthopedic Surgery, Chungnam National University School of Medicine, Daejeon 34134, Republic of Korea; 2Department of Anaesthesia, Chungnam National University Sejong Hospital, Sejong 30099, Republic of Korea; 3Department of Orthopedic Surgery, Konyang University College of Medicine, Daejeon 35365, Republic of Korea

**Keywords:** ankle fracture, anesthetic, nerve block, pain management, postoperative pain control

## Abstract

*Background and Objectives*: Patients experience severe pain after surgical correction of ankle fractures. Although their exact mechanism is unknown, dexamethasone and epinephrine increase the analgesic effect of anesthetics in peripheral nerve blocks. This study aimed to compare the postoperative pain control efficacy of peripheral nerve blocks with ropivacaine combined with dexamethasone/epinephrine and peripheral nerve blocks with only ropivacaine and added patient-controlled analgesia in patients with ankle fractures. *Materials and Methods*: This randomized, controlled prospective study included patients aged 18–70 years surgically treated for ankle fractures between December 2021 and September 2022. The patients were divided into group A (*n* = 30), wherein pain was controlled using patient-controlled analgesia after lower extremity peripheral nerve block, and group B (*n* = 30), wherein dexamethasone/epinephrine was combined with the anesthetic solution during peripheral nerve block. In both groups, ropivacaine was used as the anesthetic solution for peripheral nerve block, and this peripheral nerve block was performed just before ankle surgery for the purpose of anesthesia for surgery. Pain (visual analog scale), patient satisfaction, and side effects were assessed and compared between the two groups. *Results*: The patients’ demographic data were similar between groups. Pain scores were significantly lower in group B than in group A postoperatively. Satisfaction scores were significantly higher in group B (*p* = 0.003). There were no anesthesia-related complications in either group. *Conclusions*: Dexamethasone and epinephrine as adjuvant anesthetic solutions can effectively control pain when performing surgery using peripheral nerve blocks for patients with ankle fractures.

## 1. Introduction

Severe pain after orthopedic surgery contributes to fear of surgery and post-traumatic stress disorder [1]. Postoperative pain control in patients with lower extremity fractures is an ongoing research topic. The peripheral nerve block (PNB) has recently gained popularity as an anesthetic technique for lower extremity surgery.

Ropivacaine is usually used as an anesthetic solution for PNB. The analgesic effect of PNBs with ropivacaine is brief, generally lasting <24 h postoperatively [2]. Dexamethasone and epinephrine increase anesthetic effects and may also be effective in PNBs [3]. However, few studies have investigated the usefulness of combining dexamethasone and epinephrine for PNBs [4,5,6,7]. To our knowledge, no comparative studies on pain control using conventional PNB for ankle fractures are available.

We hypothesized that PNB with dexamethasone and epinephrine is more effective than other pain control methods after conventional PNB. Therefore, this study aimed to prospectively compare PNB combined with dexamethasone/epinephrine and patient-controlled analgesia (PCA) using ketorolac after PNB anesthesia in patients with ankle fractures.

## 2. Materials and Methods

### 2.1. Patients

This single-center randomized, controlled prospective study enrolled patients aged 18–70 years surgically treated for ankle fractures between December 2021 and September 2022. Unilateral open reduction and internal fixation for ankle fractures were performed on the patients by a single surgeon who has been operating in this field for >10 years. Fracture types included fractures involving the articular surface of the distal tibia and fibula, including simple fibula fractures, bimalleolar fractures, trimalleolar fractures, and pilon fractures. Patients were blinded to their group assignment and hospitalized for at least three days for postoperative pain control.

The exclusion criteria were contraindication for PNBs; uncontrolled diabetes mellitus, peripheral vascular disease, renal or hepatic disease, or any neurologic disease; and contraindication for regional anesthesia (coagulopathy or injection site infection). Patients with body mass index <18.5 kg/m^2^, which is considered underweight according to the World Health Organization standard [8], were excluded for anesthesia safety. Patients with suspected nerve injuries or nerve injuries requiring careful postoperative observation and those at risk of compartment syndrome were also excluded (Figure 1).

This study was approved by the institutional review board (CNUSH 2021-11-003) and was conducted in accordance with the Declaration of Helsinki. This study was registered with the ISRCTN registry (ISRCTN17431025). Written informed consent was obtained from all patients. A single researcher explained and conducted the study. Patients understood, and agreed to be hospitalized for more than three days according to their will, which is common in this research institution for hospitalization of more than three days after ankle fracture surgery, and is related to the medical system. All authors approved the final version of this manuscript.

### 2.2. Study Design

All patients were anesthetized using ultrasound-guided PNB with ropivacaine. We randomly allocated 60 participants (1:1) into two groups using blinded randomization blocks. Group A received PCA with ketorolac for postoperative pain management after PNB. Group B received PCA with normal saline; instead, dexamethasone and epinephrine were added to ropivacaine during PNB. The allocation sequence was concealed from the researchers and participants in sequentially numbered, opaque sealed envelopes. The envelopes were opened only for the researchers after the enrolled participants had completed all baseline assessments, when it was time to perform the intervention in the operating room. A sample size of 59 patients was determined based on the following parameters: significance level (5%), statistical power (90%), sample ratio (1:1), variance (2.5), and difference between the two groups (1.5). To obtain a 1:1 ratio between groups, we included 60 cases (30 in each group).

In group A, PCA was initiated approximately 10 h after PNB induction [2,9]. The treatment comprised 4 mL ketorolac (120 mg) and 100 mL normal saline. An initial bolus of 8 mL was injected, followed by an additional 96 mL slowly administered with a PCA instrument (Auto Selector; Tecnica Scientifica Service, Torino, Italy) over 48 h. A maintenance dose of 2 mL/h was administered, with each additional PCA bolus containing 1 mL and a lockout interval of 15 min.

In group B, PNB was performed using an anesthetic solution of ropivacaine (Naropin^®^, AstraZeneca AB, Sodertalje, Sweden) combined with dexamethasone disodium phosphate 5 mg (5 mg/mL, Daewon Pharm. Co., Ltd., Seoul, Republic of Korea) and epinephrine 0.1 mg (1 mg/mL, Daihan Pharm. Co., Ltd., Seoul, Republic of Korea); epinephrine was added in a ratio of 1:200,000. The same PCA instrument was also used for all patients in group B. However, only normal saline was administered in the same way as in group A. We kept all patients unaware of which group they belonged to until the end of the study. To do so, the same PCA instrument was applied to all patients included in this study. In both groups, patients with visual analog scale (VAS) scores ≥ 5 received intravenous acetaminophen (Kabi paracetamol 100 mL, 1 mg/mL, Fresenius Kabi, Friedberg, Germany) for rescue analgesia. VAS scores obtained within 8 h of intravenous acetaminophen injection were excluded from the analysis. No other pain control medications or methods were used in either group.

Pain intensity (VAS score: 0, no pain; 10, worst pain imaginable) was compared between the two groups at 6, 12, 18, 24, 32, 40, 48, and 60 h after PNB. The time at which the sensation began (analgesia time) and the time at which motor function was restored were recorded. After three days of administering pain control, a questionnaire was completed to assess patients’ satisfaction with the pain control method (Likert scale). The clinical researcher collected the questionnaires and confirmed that the patients had filled them correctly (Figure 2). The clinical researcher, blinded to group allocation and not involved in the block procedure, investigated the other surgical and anesthetic data. The number of additional analgesic doses required during the same 60 h period and complications were measured by a surgeon.

### 2.3. Anesthetic and Operative Procedures

The surgeon administered the anesthetic solution (30 mL 0.75% ropivacaine) via a 50-mL syringe connected to a venous catheter with a 100-mm, 23-gauge spinal needle. A registered nurse prepared the anesthetic solution by adding dexamethasone and epinephrine to the ropivacaine (Figure 3).

A standard noninvasive monitor was used in the operating room, and an intravenous line was secured. A 3–12 MHz linear transducer with an ultrasonic device (LOGIQ S7; GE Healthcare, Seoul, Republic of Korea) was used. All patients received an ultrasound-guided single-injection sciatic nerve block at the mid-thigh level on the lateral side, and a femoral nerve block in the inguinal area. The femoral and sciatic nerves were each injected with 15 mL of the solution under aseptic conditions.

The patients were placed in a supine position, and a femoral nerve block was performed. A spinal needle was inserted via an in-plane approach while ultrasonographically visualizing the short axis of the femoral nerve at the inguinal level. Ultrasound visualization helped confirm the correct needle position. Following this, a negative aspiration was performed and 15 mL of local anesthesia was injected in fractionated doses over a minute. A sciatic nerve block was performed in the proximal 15 cm of the popliteal fossa before separating the tibial and peroneal nerves, while the patient was in a supine position with their knee flexed at 60 degrees. Local anesthesia injection around the sciatic nerve was performed with the same technique employed for the femoral nerve block (Figure 4).

After confirming the loss of a cold sensation in the ankle and lower leg, surgery was started. The same orthopedic team conducted all surgeries. At surgery commencement, patients complaining of mild pain in the medial malleolar area received 5 mL local injections of 1% lidocaine hydrochloride (20 mg/mL, Daihan Pharm. Co., Ltd., Seoul, Republic of Korea) for analgesia. A tourniquet was applied to the distal thigh in all patients. In some patients, thigh pain during the surgery was relieved by removing the tourniquet after a rubber bandage was wrapped around the proximal tibia. A scrub nurse recorded the operation start and end times.

### 2.4. Statistical Analyses

All statistical analyses were performed using SPSS software (v.24.0; IBM Corp., Armonk, NY, USA). VAS and satisfaction scores were compared between pain control methods using a Mann–Whitney U test. Sex and diagnoses were compared using a chi-square or Fisher’s exact test. Operation time and motor and sensory function recovery times were compared between the groups using an independent t-test. Results are considered statistically significant at *p* ≤ 0.05.

## 3. Results

All 60 enrolled patients completed the study. Patient characteristics were similar between the two groups. Group A included 20 men, and Group B comprised 19 men. The fracture types in both groups are shown in Table 1.

The anesthesia procedure and operation times were similar between the groups. It took an average of 14.4 and 13.4 min to perform the skin incision in groups A and B, respectively. The incision was delayed by approximately 5 min if there was insufficient anesthesia. In three group A and two group B patients, 1% lidocaine (5 mL) was locally injected into the surgical site at the start of the operation due to medial malleolar pain. The average time between the groups from anesthesia initiation to surgery completion was similar. After surgery, the surgeon confirmed that all patients’ ankles and lower extremities were completely paralyzed. Analgesia duration and the time of recovery of motor function after surgery differed significantly between the groups. In group A, the analgesia time was an average of 11.6 h (8–14), and motor function recovery began at an average of 12.0 h (8–15.5). In group B, sensation was restored at 35.8 h (20–42), and movement at 35.6 h (21–41) (*p* < 0.001). There were no complaints of pain in both groups at 6 h. In group A, sensation began to return before PCA was applied in some cases, at approximately 10 h, but no cases of severe pain were reported until the beginning of PCA. If additional pain control was required postoperatively, intravenous acetaminophen was administered. In group A, five patients received intravenous acetaminophen between 12 and 18 h, and one had an additional same dose injection. In group B, three patients received intravenous acetaminophen, two between 24 and 32 h and one between 32 and 40 h. Satisfaction scores (Likert scale) differed significantly between the groups (group A: 7.3; group B: 8.5; *p* = 0.003). Complications related to anesthesia or surgery were not reported in any group (Table 2).

At 12, 18, 24, 40, 48, and 60 h after surgery, group B had significantly lower VAS pain scores (*p* < 0.001, <0.001, <0.001, 0.007, 0.001, 0.001, respectively); no significant difference was noted at 6 and 32 h (*p* = 1.000 and 0.082, respectively) (Figure 5, Table 3).

## 4. Discussion

The combined use of ropivacaine with dexamethasone and epinephrine for PNB provided significantly prolonged and better analgesic effects than PCA after PNB anesthesia with ropivacaine alone. Moreover, patients who received dexamethasone and epinephrine experienced significantly less pain than the controls, even after the anesthetic effect had worn off. This finding may be related to the drug effect of dexamethasone or epinephrine, or a psychological effect related to delayed pain onset after anesthesia. Two patients in the dexamethasone and epinephrine group reported VAS pain scores of 0 throughout the investigation, even after recovering motor and sensory nerve function.

However, no significant difference was noted in VAS scores between the two groups at 32 h after surgery. In group A, patients complained of peak pain about 24 h after surgery, which gradually decreased. However, patients in group B complained of peak pain at around 32 h, when the effect of PNB disappeared; then, the pain level decreased further over time. Therefore, the pain difference between the two groups was considered to be the smallest at around 32 h. This finding is thought to be associated with some rebound pain, with complaints of high repulsive pain in some cases at the point at which the effects of PNB disappear [7].

In orthopedics, morphine, non-steroidal anti-inflammatory drugs (NSAIDs), and acetaminophen are common postoperative pain control medications, but they all have limitations. Patients may experience severe pain after surgery, but morphine often has side effects and can be fatal when overused [10]. NSAIDs and acetaminophen are safer than morphine, but oral administration is not ideal, given the possibility of hepatotoxicity and nephrotoxicity [11].

Ropivacaine is a long-acting amide local anesthetic. The major concern when using high doses of local anesthetic is systemic toxicity, which develops when the free serum concentration of ropivacaine exceeds the toxic threshold [12]. Ropivacaine is relatively long-acting and safe compared to other anesthetics, and is often used for PNBs.

There are conflicting reports on whether adding epinephrine to ropivacaine is effective. Most studies showing its effectiveness found that it reduced the risk of toxicity by slowing systemic absorption. However, the studies did not identify a significant effect on analgesic duration. Several researchers advocate the addition of epinephrine to large doses of local anesthetics to reduce the maximum plasma concentration [13]. Adding epinephrine to reduce the maximum plasma concentration induces local vasoconstriction at the injection site [14], thereby slowing absorption. Several studies have reported decreased Cmax and increased Tmax when adding epinephrine to ropivacaine for epidural [7], caudal [15], or regional [16] (thoracic paravertebral block) anesthesia. Conversely, for the perivascular subclavian block, Hickey et al. [17] found no effect on pharmacokinetics (Cmax, Tmax, or area under the curve) after adding epinephrine to ropivacaine.

The glucocorticoid, dexamethasone, appeared to be effective in one preclinical [18] and several clinical [4,19,20] studies. The mechanism by which dexamethasone prolongs regional anesthesia is debatable. As steroids induce vasoconstriction, one theory holds that the drug acts by reducing local anesthetic absorption [21]. A more plausible theory states that dexamethasone increases inhibitory potassium channel activity on nociceptive C-fibers (via glucocorticoid receptors), thereby decreasing their activity [22]. Most studies have reported that dexamethasone enhances the analgesic effect [4,6,23]; however, accurate dosage information is unavailable.

Given the conflicting reports of the pharmacokinetic effects of the addition of dexamethasone or epinephrine to PNBs [7,24,25,26], we investigated the results of adding a small amount of both drugs when performing PNBs with ropivacaine, with promising results.

During our study, we considered the possibility of anesthetic drug toxicity [13,27]. First, individuals with a low body mass index (<18.5 kg/m^2^) and those with systemic diseases were excluded. Second, we were careful not to cause toxicity reactions while administering the drugs to the patients. The authors restricted the use of additional drugs other than those approved in this study for patients. Additionally, 1% lidocaine administered during the surgery was minimized to 5 mL because of the aforementioned problems.

As expected, pain scores remained significantly different, with a block duration at most time points. The total number of additional analgesic injections administered over the first 60 h differed significantly between the groups. No significant complications occurred in either group; no neurological symptoms were reported.

In cases in which sensation remained on the medial side after anesthesia induction, a part of the posterior division of the femoral nerve was not completely anesthetized [28]. Additionally, since only ropivacaine was used, the anesthesia onset time was longer than in cases wherein lidocaine was used [2,9]. In such cases, additional anesthesia with 1% lidocaine was required for the superficial layer. Considering the duration of the action of lidocaine, it did not contribute to the postoperative VAS scores in our study.

In our study, both sensation and movement returned gradually. In group A, sensation returned first, whereas in group B, movement returned first. The sensation was determined based on when patients reported starting to feel pain. However, because the pain increased gradually, patients had difficulty specifying the point at which sensation was restored, and recovery of movement was often only identified after the patient had started moving their foot to some extent, which can be difficult to measure and can produce inaccuracies. In addition, accurate measurements were difficult when sensation and movement were recovered late at night or while the patient was sleeping. In some cases, the exact time point could not be identified. Therefore, investigating the exact time of motor and sensory function recovery was difficult in this study.

Most patients with ankle fractures can be discharged immediately after surgery. Administration of dexamethasone and epinephrine with PNB may be useful in cases wherein patients with fractures require or want hospitalization. Moreover, when managing an ankle fracture, we thought this method could be beneficial for pain control at home after hospital discharge, even in non-hospitalized patients. However, delayed motor and sensory function recovery caused discomfort in some patients. The transfer process at discharge, motor and sensory function recovery progress after discharge, and related precautions should be sufficiently explained. Alternatively, a saphenous nerve block in the adductor canal, rather than a femoral block, can also reduce the above problems. An ankle block can be convenient and require little concern for patient safety in some cases.

In addition, postoperative sensory loss can cause problems in terms of detecting if or when compartment syndrome occurs. We must be aware of the possibility of postoperative compartment syndrome and prepare diagnostic methods, because the period without motor and sensory functions lasts very long after PNB.

Normal pressure within a compartment is less than 10 mmHg. An intra-compartmental pressure greater than 30 mmHg indicates acute compartment syndrome and the need for fasciotomy. However, a single normal intra-compartmental pressure reading does not exclude acute compartment syndrome. Intra-compartmental pressure should be monitored serially or continuously. Several methods can be used for monitoring. Before the 1960s, most measurements of intra-compartmental pressure used needles to inject saline. When intra-compartmental pressure increases to within 10 mmHg to 30 mmHg of the patient’s diastolic blood pressure, it indicates inadequate perfusion and relative ischemia of the involved extremity. A Stryker Intra-Compartmental Pressure Monitor System (Stryker Orthopaedics, Mahwah, NJ, USA) has been used recently to evaluate compartment pressure. It is a portable monitor that uses a side port needle, a disposable syringe of saline flush, and a digital read-out manometer to allow for simple measurement of compartment pressure [29].

If there is even a slight suspicion of compartment syndrome after PNB, one should be ready to apply these methods. This may reduce the risk to some extent. However, according to research, even this can often be inaccurate and insufficient [30]. In conclusion, when compartment syndrome occurs after surgery with PNBs, it may be more difficult to diagnose than when surgery is performed with a usual anesthetic method. Therefore, anesthesia and pain control using a PNB should be applied to patients with a low risk of developing compartment syndrome, such as low-energy single and bimalleolar fractures. In patients with severe soft tissue damage or preoperative swelling, a PNB method should preferably be avoided.

It is important to note that complications may occur when applying PNB to patients and using anesthetic drugs. When high-dose local anesthetics are used, the main problem is systemic toxicity when the free ropivacaine serum concentration exceeds the threshold. However, when anesthetics are mixed with dexamethasone and epinephrine, they cause local vasoconstriction near the injection site, which slows the drug absorption, thus lowering the maximum plasma concentration of the free ropivacaine [14,21]. Thus, the combination of dexamethasone and epinephrine is considered to have a less adverse effect on systemic drug toxicity when using ropivacaine. However, one should always pay attention and be careful of the complications of drug interactions. In our study, no systemic toxicity or complications occurred. However, after using anesthetics mixed with dexamethasone during PNB in another clinical study [6], complaints of numbness and tingling sensations in the innervation area for two weeks were reported, although without statistical significance. Therefore, caution is required when using the drug, and anesthetics should be used after a detailed investigation of the patient’s medical and drug history and systemic condition.

This study is considered a well-compared study, with no dropouts of patients and measurement errors. However, this study had limitations. First, our study only included a few patients. Second, we did not perform mid- to long-term observations; thus, longer-term complications and sequelae should be studied in larger study populations after using the combination of medicines. Further research is needed to determine whether this medication combination will be effective for other types of surgery, or if this combination can be applied to nerves in other body parts. In our study, we used ropivacaine with dexamethasone and epinephrine; hence, it was unknown which of the two drugs played a major role in sustaining the anesthetic effect. In addition, it was impossible to provide the dose of the drug combination appropriate for a single use of epinephrine or dexamethasone. Moreover, future comparative studies should determine the effect of adding only dexamethasone or epinephrine to ropivacaine, and the optimal concentration of these drugs for achieving the maximum effect while reducing side effects.

## 5. Conclusions

PNB is a useful anesthetic method for patients with an ankle fracture. The adjuvant use of dexamethasone and epinephrine as anesthetic agents had an excellent effect on pain control by extending the duration of the anesthetic effect. The anesthesia method described herein could be useful if surgery is selected after carefully considering the exclusion criteria.

## Figures and Tables

**Figure 1 medicina-59-01302-f001:**
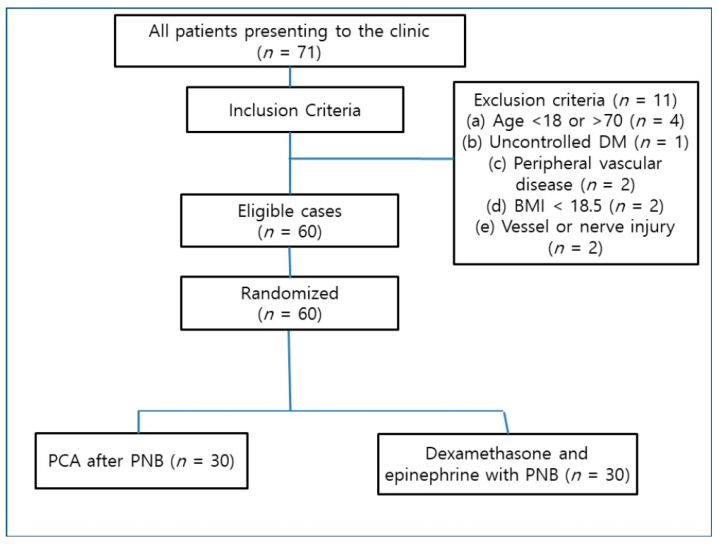
CONSORT (Consolidated Standards of Reporting Trials) flow diagram. Abbreviations: DM; diabetes mellitus, BMI; body mass index, PCA; patient-controlled analgesia, PNB; peripheral nerve block.

**Figure 2 medicina-59-01302-f002:**
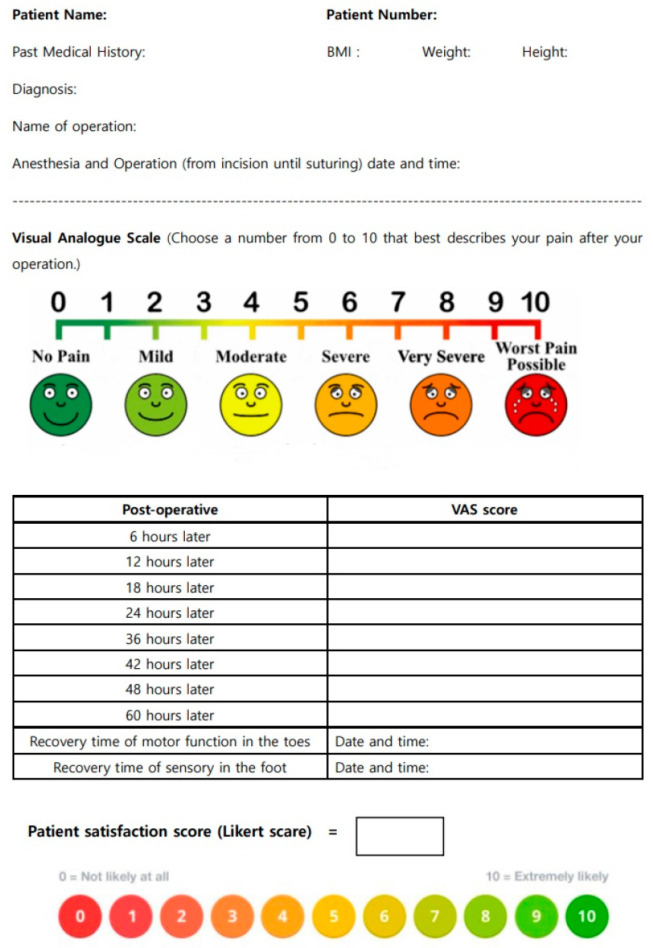
The questionnaires. Abbreviation: BMI; body mass index.

**Figure 3 medicina-59-01302-f003:**
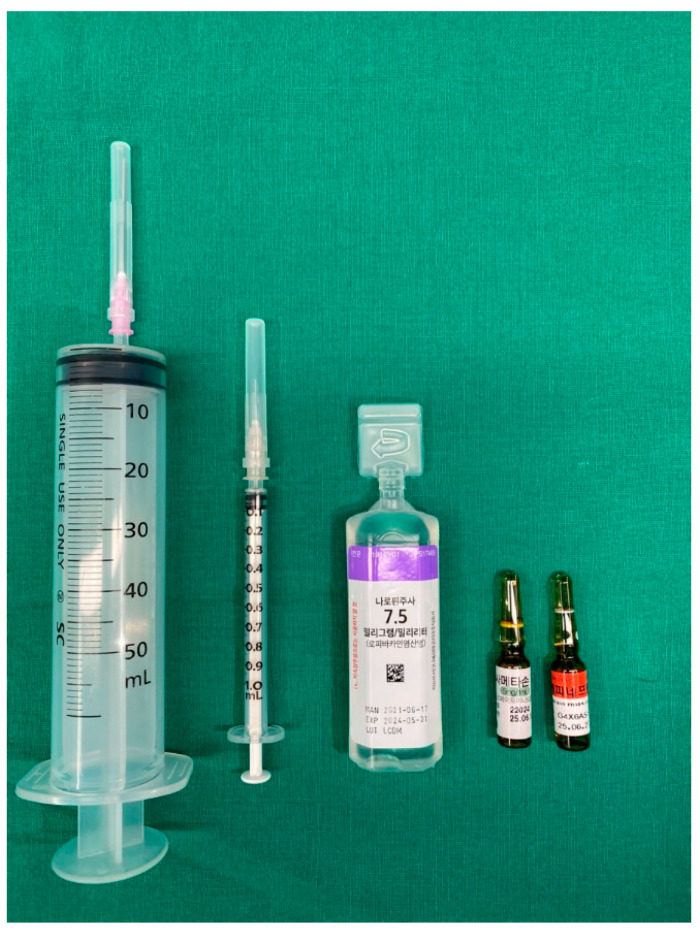
Preparations for peripheral nerve block: anesthetic drugs (ropivacaine) with dexamethasone, epinephrine, tools for mixing, and tools prepared for injection.

**Figure 4 medicina-59-01302-f004:**
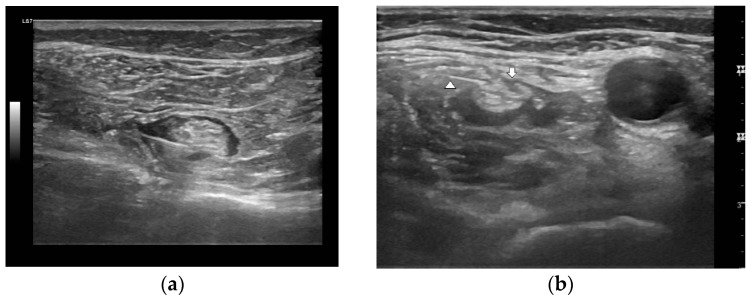
Ultrasound images. (**a**) Anesthetic fluid is injected around the sciatic nerve epineurium, and the needle tip position is confirmed via ultrasound. (**b**) The appearance following injection around the femoral nerve. An image showing the injection of anesthetics near the femoral nerve (arrow) and the needle tip (arrowhead) position is confirmed via ultrasound.

**Figure 5 medicina-59-01302-f005:**
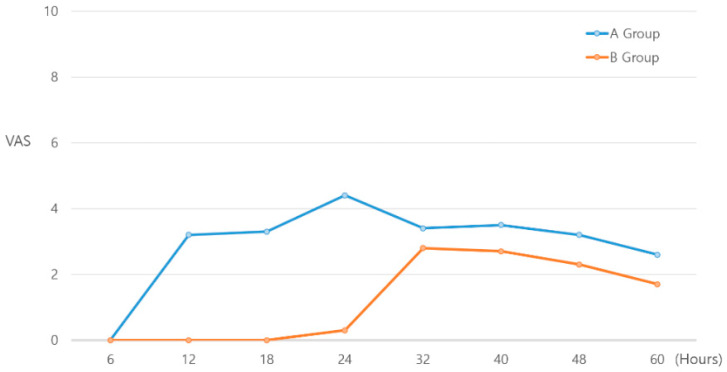
Postoperative visual analog scale (VAS) pain scores for both groups. VAS scores are significantly different between the two groups most of the time. After 32 h, similar patterns are observed between the two groups. Group A: patient-controlled analgesia after lower extremity peripheral nerve block; group B: dexamethasone/epinephrine combined with an anesthetic solution during peripheral nerve block.

**Table 1 medicina-59-01302-t001:** Patient demographic data (*n* = 60).

	Group A (*n* = 30)	Group B (*n* = 30)	*p*-Value
Age at surgery, years	35.5 ± 15.2 (19–70)	48 ± 14.9 (20–70)	0.002 *
Sex, male	20	19	0.787
BMI (kg/m^2^)	24.2 ± 2.3	24.9 ± 2.4	0.234
Affected ankle, right	16	17	1.000
Fracture type			0.808
Unimalleolar	9	9	
Bimalleolar	12	9	
Trimalleolar	7	9	
Pilon	2	3	

Abbreviation: BMI, body mass index. * Significant difference (*p* < 0.05). The statistical analysis method is as follows. Age was analyzed using a Mann–Whitney test, sex with a chi-square test, BMI with an independent *t*-test, and fracture type Fisher’s exact test.

**Table 2 medicina-59-01302-t002:** Anesthesia and surgical outcomes in both groups.

Case	Group A (*n* = 30)	Group B (*n* = 30)	*p*-Value
Operation time (min)	54.8 ± 16.7	51.4 ± 17.8	0.448
Time from anesthesia to start of surgery (min)	14.37 ± 4.20	13.40 ± 4.04	0.458
Additional injections (*n*)	5	3	0.704
Analgesia time (h)	11.6 ± 2.3	35.8 ± 8.3	<0.001 *
Motor block time (h)	12 ± 2.5	35.6 ± 7.0	<0.001 *
Likert scale	7.3 ± 1.8	8.5 ± 1.2	0.003 *

* Significant difference (*p* < 0.05). An independent *t*-test and Mann–Whitney test were used for analysis.

**Table 3 medicina-59-01302-t003:** Postoperative VAS pain score in both groups, stratified by treatment.

Postoperative Hour	VAS Pain Score		
	Group A	Group B	*p*-Value
6	0 ± 0	0 ± 0	1.000
12	3.2 ± 1.1	0 ± 0	<0.001 *
18	3.3 ± 0.9	0 ± 0	<0.001 *
24	4.4 ± 1	0.3 ± 1.0	<0.001 *
32	3.4 ± 0.9	2.8 ± 1.6	0.082
40	3.5 ± 0.9	2.7 ± 1.5	0.007 *
48	3.2 ± 0.8	2.3 ± 1.2	0.001 *
60	2.6 ± 1.0	1.7 ± 0.8	0.001 *

Abbreviation: VAS, visual analog scale. * Significant difference (*p* < 0.05). A Mann–Whitney test was used for analysis.

## Data Availability

The data presented in this study are available on request from the corresponding author.

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
