# Peer review of "A Comparative Analysis of Pain Control Methods after Ankle Fracture Surgery with a Peripheral Nerve Block: A Single-Center Randomized Controlled Prospective Study"

_medicina, 2023, doi:10.3390/medicina59071302_

Round 1

Reviewer 1 Report

Some reader might be confused about the material and methods :

-Was this study comparing PCA and PNB?

-Was this study evaluating the addition of dexamethason/epinephrine to the PNB?

please clarify

Need reviewing for some paraphrases.

Author Response

Thank you so much. 

Reviewer 2 Report

51. I was unclear about what types of fractures were included until seeing Table 1.  Perhaps you should either expand on fracture type here, or reference Table 1 in the text.

52. Is it common to hospitalise patients for three days after ankle fracture fixation at your institution? This is generally an outpatient procedure for single and bimalleolar fractures. Is this your usual care, or were patients kept in hospital longer specifically for research purposes?

71. I'm unclear whether your randomisation occurred sequentially or in blocks.  It would be helpful to clarify whether each successive patient was individually randomised.

85. What is the justification for using a placebo PCA in Group B?  This seems like a strange decision to me, and should be addressed in the text.  Using identical PCA solutions would have allowed you to meaningfully compare doses used between the two groups, which is a useful adjunct measure of pain levels and could conceivably have strengthened your results.

90. Were any patients provided with ice or other local cryotherapy for analgesia?

249. This advice is entirely inappropriate.  Skin pallor and pulselessness are very late findings of compartment syndrome and should not be used for diagnosis. If you're going to discuss diagnostic strategies, you should include measurement of compartment pressures with a Stryker pen or equivalent device, with comparison to diastolic blood pressure or MAP.  Undiagnosed or delays in diagnosing compartment syndrome is the biggest risk of using PNB for distal tibia fractures.  My feeling is that you should be very clear in your discussion regarding this risk, and I would advise you to strongly recommend against the use of this technique in situations where there is a reasonable risk of compartment syndrome.  Low-energy single and bimalleolar fractures are low risk.

Author Response

Thank you so much.

Round 2

Reviewer 1 Report

The comments have been addressed well